# Fuzzy Learning Machine

**Junbiao Cui**[1]
945546899@qq.com

**Jiye Liang**[1]*
ljy@sxu.edu.cn

[1] Key Laboratory of Computational Intelligence and Chinese Information Processing of Ministry of Education, School of Computer and Information Technology, Shanxi University, Taiyuan 030006, Shanxi, China.

## Abstract

Classification is one of the most important problems in machine learning and the nature of it is concept cognition. So far, dozens of different classifiers have been designed. Although their working mechanisms vary widely, few of them fully consider concept cognition. In this paper, a new learning machine, fuzzy learning machine (FLM), is proposed from the perspective of concept cognition. Inspired by cognitive science, its working mechanism is of strong interpretability. At the same time, FLM roots in set theory and fuzzy set theory, so FLM has a solid mathematical foundation. The systematic experimental results on a large number of data sets show that FLM can achieve excellent performance, even with the simple implementation.

## 1 Introduction

Given an input space $\mathcal{X}$, an output space (a finite set) $\mathcal{Y}$, and an unknown function $\varphi : \mathcal{X} \to \mathcal{Y}$, the goal of classification is finding the $\varphi$ or an approximation to it. Classification is one of the most important problems in machine learning, dating back to the origins of machine learning. In classification, each member of the output space $\mathcal{Y}$ corresponds to a concept. In essence, the process of classification is the process of concept cognition. Concept contains our knowledge about the world, and we use concept to understand and organize the world. Without it, there will be no human intelligence at all. That is why classification plays an important role in machine learning even in artificial intelligence.

Over the past decades, a large number of classifiers have been designed and achieved success on different tasks. The literature [1] systematically summarizes 179 classifiers from 17 different families, which contains almost all existing classifiers. Usually, generalization performance is regarded as an important dimension for evaluating classifiers. With the increasing depth and breadth of machine learning applications, classifiers are expected to possess more and more good properties, such as interpretability [2, 3, 4], robustness [5, 6, 7], and so on.

The existing classifiers follow different design principles (see **Appendix** A.2.1 for detailed analysis), but they are designed without full consideration for the nature of classification and how humans do it. For a specific scenario, the existing classifiers often perform well on one evaluation dimension but poorly on others. For example, deep neural network has exceed the human level in very specific settings [8]. Unfortunately, its working mechanism is difficult to be understood due to the complexity of the network structure [9, 10], and it is also very vulnerable to be attacked [11, 12]. Instead, human can perform well on a variety of tasks and on different evaluation dimensions.

To implement a classifier that is easy to understand and interpretable, one effective approach is to draw on relevant research in cognitive science. As mentioned above, classifying is essentially the

---

*Corresponding author

process of concept cognition. In cognitive science, there is a lot of valuable theories about how human learn, represent and use concept [13]. However, few of the existing classifiers are designed based on these theories, which leads to that the existing classifiers are unable to take into account the performance on multiple evaluation dimensions. Inspires by this, we are encouraged to design a new classifier based on concept cognition, which can simultaneously perform well on different evaluation dimensions. In this paper, a new learning machine is designed based on the following principles.

**Similarity is the basis of concept representation.** Similarity is a fundamental concept within cognitive science and plays a crucial role in the process of human classification [14]. Concepts are represented based on similarity for children, which is also a basic choice for adults [15]. Based on this principle, we prove the equivalence between classification and equivalence relation (ER) via binary relation in set theory. Considering that ER is a special similarity, we actually demonstrate that all classification problems can be solved based on similarity (see **Section** 2.1).

**Concept is fuzzy.** An important conclusion in cognitive science is that fuzziness is the intrinsic property of concept [13]. The fuzziness and the randomness are the two most important sources of the uncertainty. In machine learning, probability theory is often used for dealing with the uncertainty, such as Bayesian methods [16, 17], label distribution learning [18], etc. It is sufficient for probability theory to deal with the randomness rather than the fuzziness [19]. However, the intrinsic property of concept is just the fuzziness rather than the randomness. An intuitive example is shown in Figure 1.

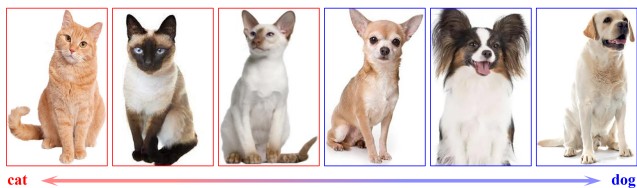

Figure 1: The visual concepts of 'cat' and 'dog' are of fuzziness. In fuzzy set theory, the membership degree of the 3rd image to 'cat' ('dog') can be $0.6$ ($0.5$). In probability theory, the probability of the 3rd image being 'cat' ('dog') can be $0.6$ ($1 - 0.6 = 0.4$). Obviously, the latter is unreasonable, because the 3rd image can not be 'cat' in some case and be 'dog' in the other case.

Based on this principle, fuzzy equivalence relation (FER) in fuzzy set theory [20] rather than ER is chosen as the core component of the new learning machine (see **Section** 2.2).

**Exemplar theory based concept representation** In cognitive science, there are several theories of concept representation [13]. Exemplar theory is chosen to solve concept representation problem, in this paper. Because it not only has a good self-consistency in cognitive science, but also it can be naturally integrated with the data-driven machine learning paradigm (see **Section** 2.3).

To summarize, the main contributions of this paper are four-folds as below:
(1) This paper revisits classification problem from the perspective of concept cognition, and proves the equivalence between classification problem and ER. Furthermore, considering that concept is intrinsically fuzzy, the classification problem is modeled as an FER problem.
(2) A new and general learning machine FLM is proposed based on concept cognition for solving FER problem.
(3) A new fuzziness permissible loss is designed. Given the new loss, this paper prove that FER can be approximated effectively by fuzzy similarity relation (FSR). And then a neural network based FLM (NN-FLM) is designed.
(4) Extensive experimental results demonstrate the superiority of NN-FLM on different evaluation dimensions including interpretability, robustness, and generalization performance.

## 2  Fuzzy Learning Machine

**Basic notations** Let $\mathcal{X}$ and $\mathcal{Y} = \{c_1, c_2, \cdots, c_k\}$ be the input and output spaces, respectively. Let $\varphi : \mathcal{X} \to \mathcal{Y}$ be the unknown target function and $\forall x \in \mathcal{X}$, $\varphi(x) \in \mathcal{Y}$ is the true class label of the sample $x$. The goal of classification is to approximate $\varphi$ as closely as possible. Let $D_{\text{train}} = \{(x_i, y_i) | x_i \in \mathcal{X}, y_i = \varphi(x_i) \in \mathcal{Y}, i = 1, 2, \cdots, n\}$ and $D_{\text{test}} = \{ x_j | x_j \in \mathcal{X}, j = n+1, n+2, \cdots, n+m \}$ be the training and test data sets, respectively.

## 2.1 Classification Problem Can Be Solved Based on Similarity

Researches [14, 15] on cognitive science prove that similarity is the basis of concept representation. However, the relationship between similarity and classification (e.g. whether all classification problems can be solved by using similarity) still lacks a mathematical justification. This section will accomplish it. The mathematical tool used in this section is binary relation in set theory. See **Appendix** A.1.2 for a brief introduction.

First, the formal definition of classification problem is given.

**Definition 1** *For an input space $\mathcal{X}$, an output space $\mathcal{Y} = \{c_1, c_2, \cdots, c_k\}$, and an unknown target function $\varphi : \mathcal{X} \to \mathcal{Y}$. The $(\mathcal{X}, \mathcal{Y}, \varphi)$-classification problem is defined as: $\forall x \in \mathcal{X}$, finding $\varphi(x)$.*

It should be pointed out that the **Definition** 1 is so general that almost all classification problems studied in machine learning can be represented by it.

**Definition 2** *Given a $(\mathcal{X}, \mathcal{Y}, \varphi)$-classification problem, (1) let $\mathcal{X} \times \mathcal{X}$ be the new input space, (2) let $\{0, 1\}$ be the new output space, (3) let $\varphi^\dagger : \mathcal{X} \times \mathcal{X} \to \{0, 1\}$ be the new target function, where $\forall (x_i, x_j) \in \mathcal{X} \times \mathcal{X}, \varphi^\dagger((x_i, x_j)) = \mathbb{I}(\varphi(x_i) = \varphi(x_j))$, then finding the $\varphi^\dagger$ is defined as its adjoint $(\mathcal{X} \times \mathcal{X}, \{0, 1\}, \varphi^\dagger)$-classification problem.*
*Obviously, the $\varphi^\dagger$ is an equivalence relation (ER) on $\mathcal{X}$ (see **Definition** 7 in **Appendix** A.1.2) and the adjoint $(\mathcal{X} \times \mathcal{X}, \{0, 1\}, \varphi^\dagger)$-classification problem is a binary-classification problem.*

Based on the above definitions, we have the following proposition.

**Proposition 1** *The $(\mathcal{X}, \mathcal{Y}, \varphi)$-classification problem is equivalent to its adjoint $(\mathcal{X} \times \mathcal{X}, \{0, 1\}, \varphi^\dagger)$-classification problem, i.e. if one problem is solved, then the other problem will also be solved. (See **Appendix** A.3.1 for proof)*

The **Proposition** 1 establishes the relationship between ER and classification problem. For any $(\mathcal{X}, \mathcal{Y}, \varphi)$-classification problem, we can always solve it equivalently by solving its adjoint $(\mathcal{X} \times \mathcal{X}, \{0, 1\}, \varphi^\dagger)$-classification problem. Because ER $\varphi^\dagger : \mathcal{X} \times \mathcal{X} \to \{0, 1\}$ is a special similarity on $\mathcal{X}$. Therefore, we can solve any classification problem by finding a appropriate similarity.

## 2.2 Capture Fuzziness of Concept

The mathematical tools used in this section are fuzzy set theory and binary fuzzy relation. See **Appendix** A.1.3 for a brief introduction.

A large number of theories and experiments in cognitive science have shown that fuzziness is an intrinsic property of concepts [13]. A short mathematical argument is as follows. For any $(\mathcal{X}, \mathcal{Y}, \varphi)$-classification problem, we have $|\mathcal{X}| > |\mathcal{Y}|$. So $\exists x_i, x_j \in \mathcal{X}, x_i \neq x_j$, such that $\varphi(x_i) = \varphi(x_j)$. The membership degrees of $x_i$ and $x_j$ to the concept $\varphi(x_i)$ are equal even if $x_i$ and $x_j$ may be very different. When fuzziness of concept is introduced, the dilemma can be solved directly. Because $x_i$ and $x_j$ are different, their membership degrees to the concept $\varphi(x_i)$ are different. As shown in Figure 1, the membership degree of the 1st image to 'cat' is greater than the 3rd image, and the membership degree of 3rd image to 'dog' is less than the 6th image.

The **Section** 2.1 concludes that any classification problem can be solved based on the special similarity, i.e. $\forall x_i, x_j \in \mathcal{X}$, either they belong to the same concept then $\varphi^\dagger((x_i, x_j)) = 1$, or they don't belong to the same concept then $\varphi^\dagger((x_i, x_j)) = 0$. Obviously, this assumption is hard to satisfy in real world because it does not capture the intrinsic fuzziness of concept.

Based on the above analysis, a new mathematical tool is needed to effectively deal with the fuzziness of concept. Fortunately, an effective solution is found in the fuzzy set theory that is introduced by Lotfi A. Zadeh [20] in 1965. In it, the crisp definition of the set is extended to a fuzzy one. Specifically, in set theory, the membership degree of an object to a set is crisp, i.e. either it belongs to the set then membership degree is 1, or it does not belong to the set then the membership degree is 0. Instead, in fuzzy set theory, the membership degree of an object to a set is fuzzy, which is represented by a real number in the interval $[0, 1]$ (see **Definition** 11 in **Appendix** A.1.3 and Figure 1).

To capture the fuzziness of concept, ER in **Definition** 2 is replaced by FER (see **Definition** 16 and **Theorem** 3 in **Appendix** A.1.3), which leads to the following definition.

**Definition 3** *Given a $(\mathcal{X}, \mathcal{Y}, \varphi)$-classification problem, (1) let $\mathcal{X} \times \mathcal{X}$ be the new input space, (2) let $[0,1]$ be the new output space, (3) let $\varphi^{\ddagger} : \mathcal{X} \times \mathcal{X} \to [0,1]$ be the new target function, where $\varphi^{\ddagger}$ is an FER on $\mathcal{X}$, and $\forall (x_i, x_j) \in \mathcal{X} \times \mathcal{X}$, $\varphi^{\ddagger}((x_i, x_j))$ be the degree that $x_i$ and $x_j$ belong to the same concept. Then finding the $\varphi^{\ddagger}$ is defined as its adjoint $\left(\mathcal{X} \times \mathcal{X}, [0,1], \varphi^{\ddagger}\right)$-FER problem.*

To solve the $(\mathcal{X}, \mathcal{Y}, \varphi)$-classification problem, it is more reasonable to solve the $\left(\mathcal{X} \times \mathcal{X}, [0,1], \varphi^{\ddagger}\right)$-FER problem than the $\left(\mathcal{X} \times \mathcal{X}, \{0,1\}, \varphi^{\dagger}\right)$-classification problem. Because it is fuzzy whether two samples belong to the same concept in the former. This assumption fits well with the principle that concept is fuzzy. For a learning task, solving the $\left(\mathcal{X} \times \mathcal{X}, [0,1], \varphi^{\ddagger}\right)$-FER problem is finding an FER on $\mathcal{X}$ that can approximate $\varphi$ by using $D_{\text{train}}$. The process can be formally described as follows

$$\text{f}^* = \arg \min_{\text{f}} \mathcal{L}\left(D_{\text{train}}, \text{f}\right) + \gamma \mathcal{R}(\text{f}) \ s.t. \ \text{f} \in \{\text{g}|\text{g is an FER on } \mathcal{X}\}, \tag{1}$$

where $\mathcal{L}$ is the loss function that measures how well the f fits the training data set $D_{\text{train}}$, $\mathcal{R}$ is the regularization term, and $\gamma > 0$ is the tradeoff parameter.

### 2.3 Exemplar Theory Based Concept Representation

The main knowledge used in this section is concept representation theory. See **Appendix** A.1.1 for a brief introduction.

Suppose we have obtained an FER f* (the optimal solution of formula (1)) by using $D_{\text{train}}$. From the perspective of cognitive science, we have constructed the basis used for representing all concepts in $\mathcal{Y}$. Next, we need to complete concept representation based on the basis.

In cognitive science, exemplar theory is about concept representation [13]. Its core idea is that human represents a concept by remembering some objects that belong to it. Therefore, we need to find some exemplars that can represent every concept in $\mathcal{Y}$. In cognitive science, which object is suitable as an exemplar of a concept is still an open question, but it is an unequivocal fact that the similarity is the key of exemplar theory [13]. In a data-driven learning task, the exemplar set of every concept can be determined by the following definition.

**Definition 4** *Given a $(\mathcal{X}, \mathcal{Y}, \varphi)$-classification problem and a training data set $D_{\text{train}}$. Let f* be the FER found by using $D_{\text{train}}$. $\forall c \in \mathcal{Y}$, let $X_{\text{train}}^c = \{x_i | (x_i, y_i) \in D_{\text{train}}, y_i = c\}$. And $\forall x \in X_{\text{train}}^c$, let $\mu(x,c) = \frac{1}{|X_{\text{train}}^c|} \sum_{x_i \in X_{\text{train}}^c} \text{f}^*((x, x_i))$. Then, $\forall c \in \mathcal{Y}$, the exemplar set is defined as*

$$E_c = \left\{x \,|\, x \in X_{\text{train}}^c, \mu(x,c) \text{ is the top} - n_{\text{exe}}^c \text{ largest value in } \{\mu(x_i, c) \,|\, x_i \in X_{\text{train}}^c\}\right\},$$

*where $n_{\text{exe}}^c$ is a manually specified parameter.*

The above definition is intuitive. Take Figure 1 as an example. Among the 3 images of 'cat', the 1st image is more suitable as an exemplar of concept 'cat' than the other 2 images, because the 1st image is more similar to the 3 images of 'cat' than the other 2 images.

After obtaining the exemplar set of every concept, the predicting process can be described as

$$\hat{y} = \arg \max_{c \in \mathcal{Y}} \frac{1}{|E_c|} \sum_{x_i \in E_c} \text{f}^*((x, x_i)), \ \forall x \in \mathcal{X}. \tag{2}$$

### 2.4 The Framework of FLM

Combining the **Sections** 2.1-2.3, the training and test processes of FLM are given as follows.

| **Algorithm 1:** The training process of FLM |
|---|
| **Input**: The training data set $D_{\text{train}}$, the number of exemplars of every class $n_{\text{exe}}^c, \forall c \in \mathcal{Y}$. |
| Find f* by solving optimization problem (1). |
| **for** $c \in \mathcal{Y}$ **do** |
|     Compute the exemplar set $E_c$ of class $c$ by **Definition** 4. |
| **end** |
| **Output**: f* and $E_c, \forall c \in \mathcal{Y}$. |

| **Algorithm 2:** The test process of FLM |
|---|
| **Input**: The test data set $D_{\text{test}}$, the learned f*, and the exemplar set $E_c, \forall c \in \mathcal{Y}$. |
| **for** $x_j \in D_{\text{test}}$ **do** |
|     Compute the predicting label $\hat{y}_j$ of sample $x_j$ by formula (2). |
| **end** |
| **Output**: $(x_j, \hat{y}_j), \forall x_j \in D_{\text{test}}$. |

## 2.5 Relationship to the Existing Methods

The existing classifiers [1] aim to learn a function $f : \mathcal{X} \to \mathcal{Y}$, while FLM aims to learn an FER $f_{FER} : \mathcal{X} \times \mathcal{X} \to [0,1]$. Therefore, the existing classifiers and FLM solve classification problem from different perspectives. FLM is designed based on concept cognition, while most existing classifiers seldom fully consider concept cognition. There is also a large corpus fuzzy classifiers [21] that use fuzzy set theory for classification task. Compared with most existing fuzzy classifiers, FLM is more suitable for data-driven learning tasks.

The distance metric learning [22, 23, 24], the siamese network [25, 26, 27], and the relation network based methods [28, 29, 30] both aim to learn a similarity $s : \mathcal{X} \times \mathcal{X} \to \mathbb{R}$. And FLM aims to learn an FER $f_{FER} : \mathcal{X} \times \mathcal{X} \to [0,1]$. On the one hand, the $f_{FER}$ is a special similarity. So these three methods and FLM are similar in this respect. On the other hand, the $f_{FER}$ needs to satisfy four properties (i.e. normalization, reflexivity, symmetry, and transitivity), while the $s$ usually cannot satisfy these properties at the same time. Therefore these methods can not fully capture the nature of classification.

The detailed analysis about the above conclusions can be found in **Appendix** A.2.

## 3 NN-FLM: A Neural Network Based FLM on $\mathbb{R}^d$

In this section, the most popular input space, $\mathcal{X} = \mathbb{R}^d$, is taken into account. And the goal is to instantiate optimization problem (1) and solve it. The mathematical tools used in this section are binary fuzzy relation and the transitive-closure. See **Appendix** A.1.3 for a brief introduction. And the code implementation is provided in supplementary material.

### 3.1 The Separability of Adjoint $\left(\mathbb{R}^d \times \mathbb{R}^d, \{0,1\}, \varphi^\dagger\right)$-Classification Problem

To solve the adjoint $\left(\mathbb{R}^d \times \mathbb{R}^d, \{0,1\}, \varphi^\dagger\right)$-classification problem described in the **Definition** 2, classifiers need to deal with pairs of samples on $\mathbb{R}^d$. One of the simplest methods is to concatenate the feature vectors of a pair of samples, i.e.

$$D_{\text{train}}^{\text{concat}} = \left\{ \left( [\mathbf{x}_i; \mathbf{x}_j], \mathbb{I}\left(y_i = y_j\right)\right) \mid \left(\mathbf{x}_i, y_i\right), \left(\mathbf{x}_j, y_j\right) \in D_{\text{train}} \right\}, \tag{3}$$

where $\forall \mathbf{a}, \mathbf{b} \in \mathbb{R}^d$, $[\mathbf{a}; \mathbf{b}] \in \mathbb{R}^{2d}$ denotes the concatenation of them. Then, a pair of samples can be converted into a composite sample, which can be processed by classifiers designed for $\mathbb{R}^{2d}$. However, the binary-classification problem obtained by the above method is not linearly separable. See the following theorem.

**Theorem 1** *For a $\left(\mathbb{R}^d, \mathcal{Y}, \varphi\right)$-classification problem and a training data set $D_{\text{train}}$, if the sample is defined as formula (3), then the corresponding $\left(\mathbb{R}^{2d}, \{0,1\}, \varphi^\dagger\right)$-classification problem is not linearly separable. (see **Appendix** A.3.2 for proof)*

Therefore, FLM on $\mathbb{R}^d$ must be able to handle the linear inseparability of the training set.

### 3.2 The Design of NN-FLM

To solve the original $\left(\mathbb{R}^d, \mathcal{Y}, \varphi\right)$-classification problem, an FER on $\mathbb{R}^d$ (see **Definition** 3) should be found. It is very difficult to directly obtain a binary fuzzy relation that satisfies transitivity. So an indirect method is adopted. First, construct a binary fuzzy relation that satisfies reflexivity and symmetry, i.e. FSR (see **Definition** 15 in **Appendix** A.1.3). Then convert the FSR into FER.

To fully implement FLM, the following requirements must be considered. (a) The linear inseparability of the problem (see **Theorem** 1). (b) The learned function must be an FER, i.e. the learned function is normalized, reflexive, symmetric, and transitive. (c) The loss function must consider the fuzziness of concept. (d) An effective optimizer must be designed, and it should be efficient especially for large scale training data sets.

Based on the above analysis, NN-FLM includes three components: (1) Fuzzy similarity relation network, which responds to (a) and the first 3 conditions in (b), (2) Fuzziness permissible loss, which responds to (c) and indirectly satisfies the transitivity in (b), and (3) Stochastic gradient descent based optimizer, which responds to (d). An overall design of NN-FLM is shown in Figure 2.

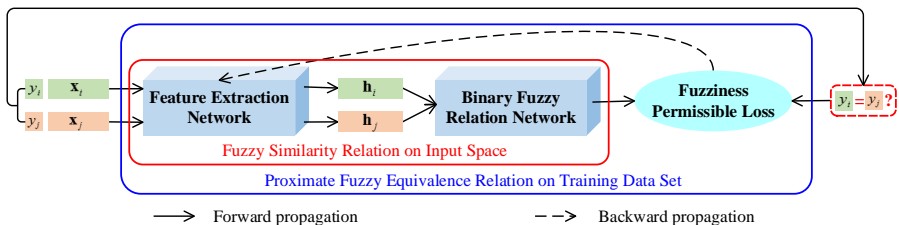

Figure 2: The overall design of NN-FLM.

**(1) Fuzzy similarity relation network** contains two parts: feature extraction network and binary fuzzy relation network.

First, the feature extraction network can be formally described as follows

$$\forall \mathbf{x} \in \mathbb{R}^d, \mathrm{h}\left(\mathbf{x}; \Theta\right) \in \mathbb{R}_+^{d_h}, \tag{4}$$

where $\mathbb{R}_+$ is the nonnegative real numbers, $d_h$ is the dimension of the latent space, and $\Theta$ is the set of learnable parameters. A multi-layer neural network with nonlinear activation function is recommended, to deal with the linear inseparability of the problem.

Second, the cosine similarity is used as the skeleton of the binary fuzzy relation network, i.e. $\forall \mathbf{h}_i, \mathbf{h}_j \in \mathbb{R}_+^{d_h}, \mathrm{g}\left(\mathbf{h}_i, \mathbf{h}_j\right) = \frac{\mathbf{h}_i^T \mathbf{h}_j}{\|\mathbf{h}_i\|_2 \|\mathbf{h}_j\|_2}$. Let $\mathrm{f}\left(\left(\mathbf{x}_i, \mathbf{x}_j\right); \Theta\right) = \mathrm{g}\left(\mathrm{h}\left(\mathbf{x}_i; \Theta\right), \mathrm{h}\left(\mathbf{x}_j; \Theta\right)\right), \forall \mathbf{x}_i, \mathbf{x}_j \in \mathbb{R}^d$, be the composite of h and g. Obviously, f is an FSR on $\mathbb{R}^d$, i.e. f is normalized, reflexive, and symmetric.

It is very difficult to make f satisfy transitivity on $\mathbb{R}^d$. However, it is easy to make f satisfy transitivity on the training sample set $X_{\text{train}} = \{\mathbf{x}_i | \left(\mathbf{x}_i, y_i\right) \in D_{\text{train}}\}$. Let $\mathbf{S} \in [0,1]^{n \times n}, s_{ij} = \mathrm{f}\left(\left(\mathbf{x}_i, \mathbf{x}_j\right)\right), \forall i, j = 1, 2, \cdots, n$, be the FSR matrix on $X_{\text{train}}$ (see **Definition** 14 in **Appendix** A.1.3). And let

$$\mathbf{T} = \mathrm{t}_{\text{FSR2FER}}\left(\mathbf{S}\right) \tag{5}$$

be the transitive-closure of $\mathbf{S}$ (see **Definition** 20, **Theorem** 4 and 5 in **Appendix** A.1.3). Then, $\mathbf{T} \in [0,1]^{n \times n}$ is an FER matrix on $X_{\text{train}}$.

**(2) Fuzziness permissible loss** For a pair of training samples $\left(\mathbf{x}_i, y_i\right), \left(\mathbf{x}_j, y_j\right) \in D_{\text{train}}$, let $t_{ij}$ be the predicting FER value of $\left(\mathbf{x}_i, \mathbf{x}_j\right)$ by formula (5). The $(\alpha, \beta)$-insensitive loss is used to measure the difference between the predicting value and the true value and can be written as

$$\mathcal{L}_{\alpha,\beta}\left(t_{ij}, y_i, y_j\right) = \begin{cases} \max\left\{t_{ij} - \alpha, 0\right\}, & y_i \neq y_j \\ \max\left\{\beta - t_{ij}, 0\right\}, & y_i = y_j \end{cases}, \tag{6}$$

where $0 \leq \alpha < 0.5 < \beta \leq 1$ are two hyper-parameters. The gap $\beta - \alpha \in (0,1]$ is used for controlling the fuzziness degree of concept. And the greater the value is, the less the fuzziness will be. Therefore, the loss on the entire training data set is

$$\mathcal{L}_{\text{FER}} = \sum_{i=1}^{n} \sum_{j=1}^{n} \mathcal{L}_{\alpha,\beta}\left(t_{ij}, y_i, y_j\right). \tag{7}$$

However, the time complexity of converting the FSR matrix $\mathbf{S}$ into the FER matrix $\mathbf{T}$ is $O\left(n^3 \log_2 n\right)$ (see **Remark** in **Appendix** A.1.3). In practice, it is not feasible to directly optimize the above loss, especially when the number of training samples $n$ is large. Fortunately, due to the intrinsic property of the transitive-closure, we can indirectly control the loss (7) by optimizing the following loss

$$\mathcal{L}_{\text{FSR}} = \sum_{i=1}^{n} \sum_{j=1}^{n} \mathcal{L}_{\alpha,\beta}\left(s_{ij}, y_i, y_j\right). \tag{8}$$

See the following theorem.

**Theorem 2** *Given a training set $D_{\text{train}} = \left\{\left(\boldsymbol{x}_i, y_i\right) | \boldsymbol{x}_i \in \mathbb{R}^d, y_i = \varphi\left(\boldsymbol{x}_i\right), i = 1, 2, \cdots, n\right\}$ and an FSR f on $X_{\text{train}} = \left\{\boldsymbol{x}_i | \left(\boldsymbol{x}_i, y_i\right) \in D_{\text{train}}\right\}$. Let $\boldsymbol{S} \in [0,1]^{n \times n}, s_{ij} = \mathrm{f}\left(\left(\boldsymbol{x}_i, \boldsymbol{x}_j\right)\right), \forall i, j = 1, 2, \cdots, n,$ be the FSR matrix on $X_{\text{train}}$, and $\boldsymbol{T} = \mathrm{t}_{\text{FSR2FER}}(\boldsymbol{S})$ be the transitive-closure of $\boldsymbol{S}$. $\forall\ 0 \leq \alpha < 0.5 < \beta \leq 1$, if $\sum_{i=1}^{n} \sum_{j=1}^{n} \mathcal{L}_{\alpha,\beta}\left(s_{ij}, y_i, y_j\right) = 0$, then $\sum_{i=1}^{n} \sum_{j=1}^{n} \mathcal{L}_{\alpha,\beta}\left(t_{ij}, y_i, y_j\right) = 0$. (see* **Appendix** *A.3.3 for proof)*

According to the **Theorem** 2, when the loss (8) reaches its minimum 0, the loss (7) also reaches its minimum 0. The loss (8) can be optimized without calculating the FER matrix on all training samples. Therefore, the loss (8) is used to complete the learning process, in this paper.

**(3) Stochastic gradient descent based optimizer** To sum up, the learning process of NN-FLM can be written as

$$\min_{\Theta} \frac{1}{n^2} \sum_{i=1}^{n} \sum_{j=1}^{n} \mathcal{L}_{\alpha,\beta} \left( f\left( (\mathbf{x}_i, \mathbf{x}_j); \Theta \right), y_i, y_j \right) + \gamma \mathcal{R}(\Theta). \tag{9}$$

The time complexity of calculating the entire FSR matrix $\mathbf{S}$ is at least $O\left(n^2\right)$, which still has a large computational burden. Fortunately, to optimize the above loss, it is not necessary to calculate the entire matrix $\mathbf{S}$ at the same time. So the above optimization problem can be solved efficiently using the stochastic gradient descent with mini-batch, even when $n$ is large.

## 4  Experiments

### 4.1  An Example on MNIST Data Set

**Experimental settings** In this section, the MNIST data set [31] is chosen to demonstrate the working mechanism of NN-FLM. In NN-FLM, a 5-layer convolutional neural network is used as the feature extraction network, and the fuzzy parameters are fixed as $\alpha = 0.2$, $\beta = 0.8$. See **Appendix** A.4.1 for more experimental details.

**Interpretability analysis** First, Figure 3a shows the FSR matrix predicted by NN-FLM on 10,000 test samples. The following observations can be seen from it. (1) The FSR value between samples from the same classes is significantly larger than that of samples from different classes. (2) The FSR matrix exhibits an obvious block-diagonal structure, where each block corresponds to a class. Combining (1) and (2), we can conclude that NN-FLM can learn a high-quality FSR from the training data. Based on the learned FSR, the concepts of 10 visual digits can be accurately represented. Second, Figure 3b shows the exemplars selected from the training set according to **Definition** 4. The following observations can be seen from it. (1) The 5 exemplars of every concept are regular, with no interrupted strokes, no distortions, well-proportioned parts of the digits, and the digits are located in the center of the image. It shows that the exemplars of the every concept selected by NN-FLM can accurately capture the characteristics of every concept. (2) There are certain changes among the 5 exemplars of every concept, which ensures that the exemplars can cover the intra-concept variations.

In summary, we can conclude that NN-FLM can learn high-quality FSR from training data. And then based on learned FSR, the high-quality exemplars can be selected to accurately represent the corresponding concept.

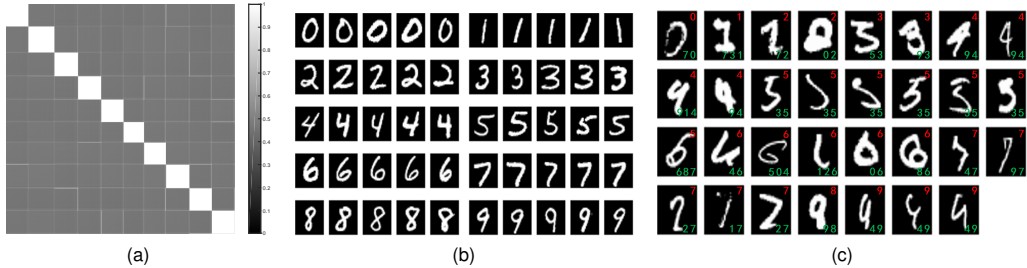

(a)  (b)  (c)

Figure 3:  (a) The predicting FSR matrix on the test set (the samples are sorted by class labels). (b) The exemplars of 10 classes selected from the training set. (c) The 31 errors of NN-FLM, with annotated label (up right) and top-2 or top-3 predicting labels (down right).

**Robustness analysis** A total of 31 test samples are misclassified and they are shown in Figure 3c. The following conclusions can be drawn from it. (1) All the misclassified samples are irregular digital images. (2) Among them, 26 images (3 images) are hit by the 2nd (3rd) class label predicted by NN-FLM. (3) For the 2 images (1st and 3rd in row 3 of Figure 3c), none of the top-3 class labels predicted by NN-FLM hits the annotated label. These 31 images are so irregular that it is difficult

to give crisp and uncontroversial labels even for humans. In this case, the class labels predicted by NN-FLM are in line with human cognition to a certain degree.

In summary, we can conclude that NN-FLM can adequately capture fuzziness of concept. Therefore, the samples with a high degree of fuzziness can be identified and assigned reasonable candidate concepts by NN-FLM. This verifies that NN-FLM is robust to the controversial samples.

## 4.2 Comparison With 179 Classifiers on 121 Data Sets

**Experimental settings** In the literature [1], 179 classifiers from 17 families (see Table 3 in **Appendix** A.4.2) are systematically compared on 121 benchmark data sets (see Table 2 in **Appendix** A.4.2). In order to verify the generalization performance of FLM, NN-FLM is chosen and compared with the above 179 classifiers. In all experiments, NN-FLM adopts a 3-layer fully connected network as the feature extraction network. The fuzzy parameters are fixed as $\alpha = 0.2, \beta = 0.8$. More details about the experimental settings can been found in **Appendix** A.4.2.

Table 1: The rank of NN-FLM among 180 classifiers on 121 data sets

| ID | Rank | ID | Rank | ID | Rank | ID | Rank | ID | Rank | ID | Rank |
|---|---|---|---|---|---|---|---|---|---|---|---|
| 1 | 1 | 21 | 1 | 41 | 1 | 61 | 2 | 81 | 10 | 101 | 21 |
| 2* | 1 | 22 | 1 | 42 | 1 | 62* | 2 | 82 | 10.5 | 102 | 23 |
| 3 | 1 | 23 | 1 | 43 | 1 | 63 | 2.5 | 83 | 11 | 103* | 23.5 |
| 4 | 1 | 24 | 1 | 44 | 1.5 | 64* | 2.5 | 84 | 12 | 104 | 24.5 |
| 5 | 1 | 25 | 1 | 45 | 1.5 | 65 | 3 | 85 | 13 | 105 | 26 |
| 6 | 1 | 26 | 1 | 46 | 1.5 | 66 | 3 | 86 | 13.5 | 106 | 26 |
| 7 | 1 | 27 | 1 | 47 | 1.5 | 67 | 3 | 87 | 14 | 107 | 29 |
| 8 | 1 | 28 | 1 | 48* | 1.5 | 68 | 3 | 88* | 14.5 | 108 | 31.5 |
| 9 | 1 | 29 | 1 | 49* | 1.5 | 69 | 3.5 | 89 | 15 | 109 | 32 |
| 10 | 1 | 30 | 1 | 50 | 1.5 | 70* | 4 | 90 | 15 | 110 | 35 |
| 11 | 1 | 31 | 1 | 51 | 1.5 | 71 | 4.5 | 91* | 15 | 111 | 36 |
| 12 | 1 | 32 | 1 | 52 | 1.5 | 72 | 5 | 92 | 15.5 | 112 | 38 |
| 13 | 1 | 33 | 1 | 53 | 1.5 | 73 | 5.5 | 93 | 15.5 | 113 | 39 |
| 14 | 1 | 34 | 1 | 54* | 2 | 74 | 6 | 94 | 16 | 114 | 39 |
| 15 | 1 | 35 | 1 | 55 | 2 | 75 | 6.5 | 95 | 16 | 115 | 39 |
| 16 | 1 | 36* | 1 | 56 | 2 | 76 | 7.5 | 96 | 16.5 | 116 | 42 |
| 17 | 1 | 37 | 1 | 57 | 2 | 77 | 8 | 97 | 17.5 | 117 | 47 |
| 18 | 1 | 38 | 1 | 58 | 2 | 78 | 9 | 98 | 18.5 | 118 | 60 |
| 19 | 1 | 39 | 1 | 59* | 2 | 79 | 9 | 99 | 19 | 119 | 62 |
| 20 | 1 | 40 | 1 | 60 | 2 | 80 | 10 | 100 | 20 | 120 | 63.5 |
| | | | | | | | | | | 121 | 65 |

(a) ID: The unique ID of every data set (see Table 2 in **Appendix** A.4.2).   (b) ∗: The 12 small sample data sets (see Table 5 in **Appendix** A.4.2).   (c) The rank considers that two or more classifiers have the same test accuracy. For example, on data set 121 there are 129 classifiers with 100% test accuracy, so the ranks of these 129 classifiers are $\left( \sum_{r=1}^{129} r \right) /129 = 65$.

**The top-10 classifiers** The 10 classifiers with the smallest mean rank among 180 classifiers on 121 data sets are selected for comparison analysis. They are NN-FLM (1st), 3 classifiers from the random forest family (2nd parRF-t, 3rd rf-t, 5th rforest-R), 3 classifiers from the support vector machine family (4th svm-C, 6th svmPoly-t, 8th svmRadCost-t), 1 classifier from the decision tree family (7th C5.0-t), and 2 classifiers from the neural network family (9th elm-kernel-m, 10th avNNet-t).

**Accuracy analysis** The test accuracy of the 179 comparison methods are from the literature [1]. We run NN-FLM on 121 data sets and record the test accuracy. Then the accuracy of 180 classifiers are used for evaluating their performance. The following conclusions can be drawn from it. (1) Due to space constraint, the accuracy of NN-FLM and other top-9 comparison methods are shown in Table 4 in **Appendix** A.4.2. And it can be seen that the mean accuracy on 121 data sets of NN-FLM is 5.49% higher than the 2nd parRF-t and 6.00% higher than the 3rd rf-t. Therefore, NN-FLM achieves a significant improvement compared with the 179 classifiers. (2) The rank of NN-FLM among 180

methods on 121 data sets is recorded in Table 1. And it can be seen that NN-FLM ranks first on 43 data sets (data sets 1-43) and ties for first place on 10 data sets (data sets 44-53), and ranks in the top-3 on more than half of the data sets (data sets 1-68). Therefore, NN-FLM achieves outstanding rank among the the 180 classifiers.

**Comparative advantage analysis** Figure 4a shows the proportion of the top-10 classifiers ranked in top-$K$ ($K = 1, 3, 5, 10, 15, 20$) among the 121 data sets. And Figure 4b shows the mean and standard deviation of the rank of the top-10 classifiers on 121 data sets. The following conclusions can be drawn from it. (1) NN-FLM ranks the top-1 on $43/121 \approx 35.5\%$ of the data sets and the top-3 on $68/121 \approx 56.2\%$ of the data sets. And NN-FLM ranks in top-20 on more that $80\%$ of the data sets. (2) None of the other 9 comparison classifiers ranks in top-20 on more than $50\%$ of the data sets. (3) NN-FLM has the smallest mean rank $10.51$. And the mean rank is significantly superior to the second-ranked classifier, parFR-t, $33.50$. (4) NN-FLM has the smallest standard deviation $14.86$. And the standard deviation is significantly lower than to the other comparison classifiers. In summary, the performance of NN-FLM is significantly superior to the comparison classifiers. At the same time, it can perform well on various tasks. In contrast, the comparison classifiers can only perform well on limited tasks. Therefore, the performance of NN-FLM is more stable than the others. The possible reason is that NN-FLM can adequately capture the nature of classification.

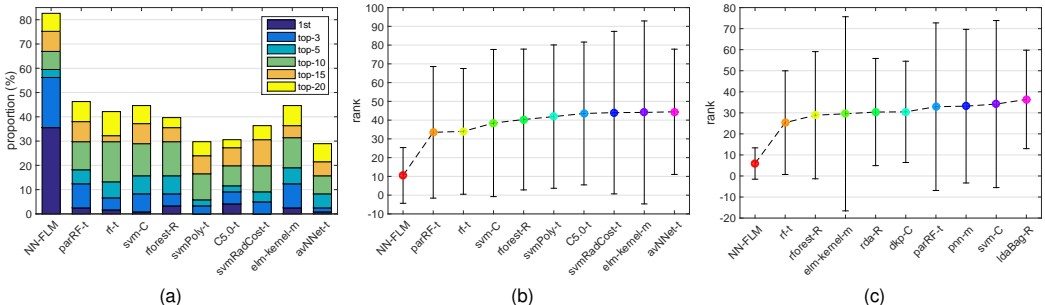

(a)         (b)         (c)

Figure 4: The statistics on top-10 methods among 180 classifiers. (a) The proportion of top-$K$ rank on 121 data sets. (b) The mean and standard deviation of the rank on 121 data sets. (c) The mean and standard deviation of the rank on 12 small data sets.

**Experiments on small sample data sets** To verify the performance of NN-FLM on small sample classification problem, we selected 12 data sets with small number of samples (see Table 5 in **Appendix** A.4.2) from the 121 data sets. In these data sets, the mean number of training samples for each class is less than 15.

We selected the top-10 methods with the smallest average rank on the 12 small sample data sets for comparison, see Figure 4c. The following conclusions can be drawn from it. (1) Compared Figures 4b with 4c, it can be seen that the top-10 classifiers on the 12 small sample data sets differ from that on the all 121 data sets. Some classifiers (svmPoly-t, C5.0-t, svmRadCost-t, and avNNet-t) rank in the top-10 on the all data sets but not on the small sample data sets. (2) The mean rank of NN-FLM on the 12 small sample data sets is still the best. (3) The rank of NN-FLM is significantly superior to the second-ranked method, fr-t ($5.91 \pm 7.44$ vs $25.33 \pm 24.62$). In summary, it can be concluded that when there are fewer training samples, the superiority of NN-FLM is more significant. The reason is that NN-FLM can capture the characteristic and intra-concept variations of concept through a small number of exemplars. Therefore, the exemplar theory based concept representation is reasonable.

## 5   Conclusion

A new learning machine based on concept cognition, fuzzy learning machine, is proposed. The new learning machine has the advantages of strong interpretability and solid mathematical foundation. And it is of certain degree of robustness because it can effectively deal with the intrinsic fuzziness in classification. At the same time, a large number of comparison experiments show that it can reach an advanced level even with a simple implementation.

## Acknowledgments and Disclosure of Funding

This work is supported by the National Key Research and Development Program of China (under grant 2020AAA0106100).

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
