# OpenReview forum: "Fuzzy Learning Machine"
_NeurIPS.cc/2022/Conference — NeurIPS 2022 Accept_

### Official Review · Reviewer_55q9 · 2022-06-29

**Rating:** 7
**Confidence:** 4
**Soundness:** 3 good
**Presentation:** 3 good
**Contribution:** 3 good

**Summary:**

This paper proposes a new learning machine for the general classification problem, which is one of the most important problems in ML/AI. The new learning machine is based on the concept cognition theory in cognitive science and fuzzy set theory in mathematics science. So its working mechanism is highly explainable and has a solid theoretical guarantee. Meanwhile, a large number of systematic experimental results demonstrate the superiority of the proposed method.

**Questions:**

(1)	The title of the manuscript is FLM, but the performance of NN-FLM is shown in experiments. Please explain the difference between FLM and NN-FLM and why only the performance of NN-FLM is analyzed.

(2)	It appears that the optimization model of NN-FLM is a general multi-layer neural network plus a special loss function. Please explain what good properties of the loss function are able to guarantee the performance of NN-FLM.

(3)	According to the descriptions in Appendix A.1, there are at least four different concept representation theories. Why choose the exemplar theory to complete the concept representation in FLM? And in some machine learning methods, ‘prototype’ is also usually used for representing a class, such as ‘means’ in ‘k-means’, ‘modes’ in ‘k-modes’, and ‘prototype ’ in ‘prototype network’. Please explain the difference between ‘prototype’ in these methods and ‘prototype’ in ‘theory theory’ and why not use prototype theory to represent the concept?


**Strengths And Weaknesses:**

The manuscript focuses on the classification problem which is one of the most important problems in ML/AI.

The manuscript re-examines the classification from the perspective of concept cognition and reveals the essence of classification. And the manuscript provides a new view to interpret the structure of the classification problem by establishing the equivalence between binary classification problem and classification problem by employing equivalence relation in set theory. Furthermore, the manuscript realizes that fuzziness of concept is the main source of uncertainty in classification and then employs the fuzzy set theory to model this kind of uncertainty.

Based on the above conclusions, the classification problem is modeled as a fuzzy equivalence relation problem, which well preserves the nature and intrinsic fuzziness of the classification problem. What’s more, the manuscript designs a clever model and loss function to approximate the fuzzy equivalence relation effectively and efficiently.

Therefore, in this manuscript, the main proposals have the theoretical basis of cognitive science, and the key conclusions are proved mathematically. And extensive experiments (compared with 179 methods on 121 data sets) verify the rationality and superiority of the proposed method.

Overall, the manuscript is clearly written and well organized with good clarity. To enhance the readability and completeness, it is suggested that some contents in the appendix should be moved to the corresponding part of the main manuscript. For example, the analysis of the working mechanism of the existing classifiers should be moved to the Introduction of the main manuscript. However, in the current manuscript, these contents are placed in Appendix A.2.

---

> ### Author Response · Authors · 2022-08-01
> **Response to Reviewer 55q9**
>
> General Response
>
> We thank the reviewer for taking the time to review our manuscript and for the valuable comments. Below is a point-by-point response to the comments.
>
> Response to Weaknesses
>
> Thanks for your advice. We have made corresponding adjustments to increase the readability in the revised manuscript (see Section 2.5).
>
> Response to Question (1)
>
> FLM is a general learning machine, which can deal with the classification problem given in Definition 1. NN-FLM is a specific implementation of FLM, which can handle the classification problem when the input space is Euclidean space. Euclidean space is the common input space in ML. However, there are still many data that is not represented by vectors in Euclidean space, such as category data. In this case, how to implement FLM is a worth research topic in the future.
>
> In addition, we take NN-FLM as the representative of FLM family to conduct comparison experiments. And the experimental results also demonstrate its effectiveness.
>
> Response to Question (2)
>
> Firstly, the goal of the optimization model is learning a similarity function. The paper argues that classification problem can be solved based on similarity, which has previously demonstrated (see Section 2.1). Therefore, the goal is reasonable.
>
> Secondly, the features of the samples may not accurately describe the intrinsic attributes of the concept and may contain information unrelated to the concept. Therefore, the excellent feature extraction ability of deep neural networks is used to learn new representations from the original features. In learned representations, the feature related to the concept are strengthened and the features unrelated to the concept are weakened. Therefore, a good FSR can be obtained, which builds a good basis for the representing the concept.
>
> Thirdly, the proposed loss function can not only capture the intrinsic fuzziness of the concept, and can be optimized efficiently. More importantly, given the proposed loss function, it is proved that FER can be approximated effectively by FSR (see Theorem 2), which preserves the nature of classification problem as much as possible.
>
> At last, exploiting the learned similarity function and the exemplar selection method, NN-FLM can select representative exemplars for each concept. And the concept can be represented effectively by selected exemplars and is used for classification.
>
> In summary, all the above factors guarantee the performance of NN-FLM.
>
> Response to Question (3)
>
> Firstly, exemplar theory is adopted to complete representation of concept. That is because exemplar theory is friendly to data-driven ML paradigm. Compared with other theory of concept representation, such classical theory, prototype theory and knowledge theory, exemplar theory hardly relies on the high-level semantic information of the features. Although the semantic information of features is unknown (This is common setting in ML), representation of concept also can be completed.
>
> Secondly, prototype in cognitive science is rely more on semantic information, for example the prototype of dogs can be represented as (four legs, hair, barking, etc.). One of the most serious problems is that no matter how you extend this representation, there are always some dogs that can not be captured by this representation. On the contrary, prototype in ML usually can be descried by a vector, and the semantic information of each dimension in vector is usually ignored. The former is of highly interpretability, but requires expert participation. The latter is the opposite. And the two are highly complementary. Therefore, the paper selects exemplar theory to complete representation of concept, which preserves the merits of the two. Specifically, concept is represented by the selected exemplars that is the samples in original space, which has good interpretability and can not cost the human efforts.

---

> > ### Comment · Reviewer_55q9 · 2022-08-07
> > **Feedback**
> >
> > Thanks for your reply. My concerns are clarified. I will keep my voting and support this paper.

---

### Official Review · Reviewer_qNtC · 2022-07-08

**Rating:** 5
**Confidence:** 4
**Soundness:** 2 fair
**Presentation:** 2 fair
**Contribution:** 2 fair

**Summary:**

In the paper "Fuzzy Learning Machine" the authors propose an approach to learn a classifier via a neural network forming a fuzzy equivalence relation. Deriving the approach from fuzzy set theory, the authors find their approach to perform particularly well across a number of datasets comparing the approach to various other classifiers.


**Questions:**

* In what regard does the proposed method really go beyond the already existing corpus of literature in fuzzy classifiers?
* Is the classification problem really solved with definition 2? To me, it seems like the information to which class at least a representative data point belongs to is missing. Hence, the problem is only solved if the y's of some representative class members are known.
* Is not it a problem that the model is not permutation invariant? At least it seems to me that it is not invariant to the order of inputs.


**Limitations:**

Limitations, except for runtime complexity to compute the FER matrix, are not really discussed. When does the approach fail and why does it fail?

**Strengths And Weaknesses:**

## Weaknesses
The idea of employing fuzzy set theory for classification tasks is not new at all and I am wondering what is now the methodological novelty of the approach. In general the idea of comparing instances / data points according to their similarity is the basic idea behind learners using kernel functions where the shape of a concept is specified via the respective kernel. However, there is also a relatively large corpus of literature on classifiers leveraging fuzzy set theory, even working exactly with neural networks and the idea of fuzzy equivalence relations. Still this related work is neither discussed nor cited in the paper. See for example the following references:
Acharya, U. Rajendra, et al. "Classification of heart rate data using artificial neural network and fuzzy equivalence relation." Pattern recognition 36.1 (2003): 61-68.
Moser, Bernhard. "On Representing and Generating Kernels by Fuzzy Equivalence Relations." Journal of machine learning research 7.12 (2006).
Meier, Andreas, and Nicolas Werro. "A fuzzy classification model for online customers." Informatica 31.2 (2007).
Senge, Robin, and Eyke Hüllermeier. "Top-down induction of fuzzy pattern trees." IEEE Transactions on Fuzzy Systems 19.2 (2010): 241-252.
Kuncheva, Ludmila. Fuzzy classifier design. Vol. 49. Springer Science & Business Media, 2000.
Sun, C-T., and J-S. Jang. "A neuro-fuzzy classifier and its applications." [Proceedings 1993] Second IEEE International Conference on Fuzzy Systems. IEEE, 1993.
Uebele, Volkmar, Shigeo Abe, and Ming-Shong Lan. "A neural-network-based fuzzy classifier." IEEE Transactions on Systems, Man, and Cybernetics 25.2 (1995): 353-361.
It is unclear to me how this part of the literature is widely ignored by the authors when they seem to come from that area.

Overall, the paper has a good structure but could benefit from proofreading. Especially, a vs an is a frequent problem in the text, e.g., "a input space", "a output space", "a FER". Then, "classifier", "concept" and "classification process"  are used without an article. Some parts also seem overly complicated to me. For example, consider the proof that a non-linear model is needed to tackle the derived problem where the instances are concatenated. I do not know whether yet another proof for the fact that an XOR problem cannot be tackled via a linear model is really needed. This could have been simplified. Furthermore, I find that the example given in Figure 1 is not very well chosen. The concepts cat and dog have crisp biological borders and a human not being able to distinguishing the two categories is rather due to epistemic uncertainty than fuzziness of the concept borders. Personally, I would also argue that non of the three cats is more or less representative of the category or concept "cat".

A claim that was made by the authors is that their approach indeed learns "concepts" instead of just assignments. However, there was no proof given in the paper that this is really the case. Especially, there is no presentation or demonstration of any particular concepts that were induced by fitting their model. I would even argue that from Figure 3 is rather becomes clear that it is learning not really any concepts as the FSR matrix shows more or less the same color for every cell not being on the main diagonal. If it was to learn real concepts I would also expect that a 0 would receive a lower membership score for the concept 1 than a 7 for example. A better overall performance is no proof for the claim that the method learns concepts.

Another branch of classification literature also tries to capture concepts for classification purposes: Analogy learning.
Bayoudh, Sabri, Laurent Miclet, and Arnaud Delhay. "Learning by Analogy: A Classification Rule for Binary and Nominal Data." IJCAI. 2007.

## Strengths
Since most people in the machine learning community will not be that much familiar with fuzzy set theory, I liked it very much that all fundamental definitions were provided by the authors in the paper or supplementary material to make it self sufficient.

According to the experiments the proposed method seems to perform very strong compared to a set of almost 200 classifiers. However, the way how the rankings were calculated is a little bit odd. Why are 65 learners sharing rank 1 with 100% accuracy receive a rank of 65? This will most likely also affect the average rank statistics compared for the ten classifiers later on. I would rather expect that performances with a tie receive the same higher rank, leaving free the next n-1 spots in the ranking.

---

> ### Author Response · Authors · 2022-08-01
> **Response to Reviewer qNtC**
>
> General Response
>
> Thank the reviewer for carefully reading and the valuable comments.
>
> The innovation of this paper not only lies in the proper use of fuzzy set theory to design classifiers, but also includes:
>
> (1)This paper demonstrates that almost all classification problems can be solved by similarity (Section 2.1).
>
> (2)This paper demonstrates that the fuzziness is unavoidable when solving classification problems (Section 2.2). Unlike most existing fuzzy classifiers (FCs), their motivation for using fuzzy set theory is more intuitive.
>
> (3)An exemplar theory based concept representation method is designed (Section 2.3).
>
> (4) A general learning machine is designed, and a specific implementation is given (Section 2.4 and 4).
>
> We apologize for the mistake that the manuscript did not contain a discussion about the existing FCs. We have added the discussions in modified version (Section 2.5 and Appendix A.2.1). The following is the responses to the comments.
>
> Response to Weaknesses
>
> 1 The focus of this paper is to demonstrate the relationship between similarity and classification problem and the fuzziness is unavoidable for solving classification problem.
>
> 2 We have added corresponding discussions (Section 2.5 and Appendix A.2).
>
> 3 They have been modified in the revised manuscript.
>
> 4 In this example, we consider the visual instead of biological sense of cats and dogs. Given only visual features, the boundary between the cat and dog is fuzzy. If the cat and dog are described in DNA features, they can be distinguished. However, the boundary between cat and dog is still fuzzy because there are a certain similarity between them in DNA features.
>
> In general, fuzziness of concept can be reduced by more accurate information and is almost impossible to be eliminated (Chapter 2 in literature 13).
>
> 5 (1) The exemplar theory is selected for concept representation. The learned FSR and the exemplars for every class are used for representing concepts.
>
> (2)From Fig 3c, it can be seen that NN-FLM selects representative exemplars for every class, which shows that NN-FLM captures the visual concepts 0-9 well.
>
> (3)In Fig 3a, the nonzero value in non diagonal indicates that NN-FLM captures the fuzziness of concepts. The Fig 3a also shows that the FSR value between 0 and 1 is lower than it between 0 and 7.
>
> (4) Analog learning can capture concepts to a certain extent. The core component of analogy learning is the 4-ary relation (Section 9.2.22 of literature 32), while the core component of the proposal is the binary relation (i.e. 2-ary relation) and this paper demonstrates that almost all classification problems can be solved by binary relation (Section 2.1). In addition, it is difficult for analog learning to deal with features that only contains low-level semantic information. The proposal can automatically extract useful features from the raw features.
>
> Response to Strength
>
> According to Friedman test, when the performance of some algorithms are the same, these algorithms will share their ranking values equally (literature [1]).
>
> Response to Questions
>
> 1 Fuzzy classifier (FC) is an important classification paradigm, which can deal with ambiguity effectively, has strong interpretability and can easily be fused with the knowledge of experts.
>
> In literature [2], the FC is defined as a classifier that uses fuzzy sets or fuzzy logic in the course of its training or operation. According to this definition, NN-FLM is a kind of FC.
>
> Compared with most existing FCs, the proposal is more suitable for data-driven machine learning tasks. Specifically, (1) it hardly relies on the semantic information of features; (2) it can automatically extract useful information for concept from low-level semantic features and capture the representation of concepts. (3) it can effectively complete training and test and can learn from large-scale data. For detailed analysis, see the Appendix A2.1 in modified version.
>
> 2 In Definition 1, the output space is a finite set. Mathematically, a finite set only needs to contain several different elements, and the meaning of each element is ignored. In this sense, classification problem can be solved with Definition 2.
>
> 3 In training stage, the f^* and E_c, \forall c \in Y are invariant to the order of training samples. In test stage, the predicting results are invariant to the order of test samples. To sum up, the proposal is invariant to the order of inputs.
>
> Response to Limitations
>
> 1 The time complexity from FSR matrix to FER matrix is discussed in Appendix A.1.2.
>
> 2 When the samples in the training set are not enough to cover all the intrinsic attributes of concepts, the learned concept representations will fail, resulting in the failure of the proposal.
>
> Reference
>
> [1] Janez Demˇsar. Statistical comparisons of classifiers over multiple data sets. Journal of Machine Learning Research, 7 (2006):1–30.
>
> [2] Ludmila1. Kuncheva. Fuzzy Classifier Design. Springer-Verlag Berlin Heidelberg GmbH, 2000.

---

> > ### Comment · Reviewer_qNtC · 2022-08-09
> > **Re: Definition Fuzzy**
> >
> > Just because two concepts share certain features does not mean the concepts are fuzzy, does it? As soon as their is one distinctive feature identifying a data point to belong to a certain concept, the border is crisp. It does not matter whether the remaining feature values are even the same. Regarding your DNA example: There are markers in the DNA for cats to actually identify them as cats and vice versa.
> >
> > However, I agree that some visuals are shared and if membership is only defined via the presence of a few visual markers the concepts may overlap in this description.

---

> > > ### Author Response · Authors · 2022-08-09
> > > **Response to Definition Fuzzy**
> > >
> > > Thanks for your advice. You are right.
> > >
> > > From a biological point of view, the concepts of “cat” and “dog” can be defined according to their DAN features. At this time, the concepts are crisp.
> > >
> > > In the field of ML, for example, in most image classification task, the goal is to learn the concepts from the images provided by data set. At this time, the learned concepts are fuzzy because the information contained in image is limited to define the concepts crisply.
> > >
> > > Thanks for your advice again. And do you have any other questions about this paper?

---

### Official Review · Reviewer_iVDZ · 2022-07-11

**Rating:** 6
**Confidence:** 4
**Soundness:** 2 fair
**Presentation:** 2 fair
**Contribution:** 3 good

**Summary:**

This paper proposes a new machine learning method for classification called Fuzzy Learning Machine. The paper draws from concepts from cognitive science to derive a method based on fuzzy similarity relations of examples on the input space. The training method learns a similarity function and selects a set of exemplars from each category used during the prediction phase to compute the similarity of new examples to the exemplars in each category and then assign it to the category with more similar examples.

**Questions:**

Below I provide a list of questions and points that could be improved in my opinion:
1. Line 16: The term "Concept cognition" lacks reference and support in the literature. If this is a term coined in the paper, then first, it should be defined what is meant by "Concept" and "Cognition" in the views of the authors, since these terms are rather vague.
2. Line 18: Why classification and not categorization? The former may be rather too specific and we do not know enough about human intelligence to pinpoint it as the approach used by it. For instance, see Jacob, Elin K.. “Classification and Categorization: A Difference that Makes a Difference.” Libr. Trends 52 (2004): 515-540. More specifically: do humans need supervision to learn?
3. Line 28: It should be made clear that DNN may exceed the human level in very specific settings.
4. Lines 29-30 and 33: The way humans do classifications is also difficult to understand and explain. Therefore, I see a conflict in pursuing human-like classifiers and classifiers that are easy to understand and interpretable at the same time.
5. Line 30: I do not see how this can be direct and efficient because we first need to understand and explain how humans do classification and, although there exist theories, this is not settled in the literature.
6. Figure 1: Please, describe where the numbers used in the legend are derived from.
7. Line 74: Test data also need labels. Otherwise, the method can not be evaluated.
8. Definition 1: Observe that "phi(x) belongs to Y" does not define any mapping in particular. Is that the intended definition? If so, it does not resemble a classification problem, and any arbitrary mapping would be a solution.
9. Line 107: I do not see why an awkward situation arises from the definitions above.
10. Equation 1: no regularization terms are considered. Is overfitting a problem?
11. I see potential problems in the way the model chooses examples for the Ec set. Consider a class with high variability, such as "vehicles". In such a class, it is hard to select k exemplars that cover well the variability of the class (cars, trucks, planes, boats) and yet are similar among all of them.
12. Line 269: "On the contrary, the class labels predicted by NN-FLM are more in line with human cognition." it is hard to conclude this from the examples shown without adequate research with different subjects and in comparison with other methods.
13. Please, provide a description of the hyperparameter tuning method used for the methods. How well were they adjusted?

Typos and writing style:
1. Line 23: possess instead of process?
2. Line 26: humans instead of human?
3. Line 42: "A concept is" or "Concepts are" instead of "Concept is" ?
4. Line 75: The question mark is not needed because the phrase is not a question.
5. Definition 1: In the notation used (X,Y, phi), phi seems rather an input than output or result. Consider something like phi = f(X,Y). (optional)
6. Line 89: why double parenthesis in ((xi, xj))?
7. Line 101: Concepts instead of Concept?
8. Line 141: "exemplar theory is about" instead of "exemplar theory is a kind of theory about".
9. Line 269: humans instead of human.


**Limitations:**

I do not see any limitations or potential negative social impact of this work.

**Strengths And Weaknesses:**

Strengths:

The method proposed is interesting and brings up a number of novelty elements. The method seems to improve significantly in relation to existing classification methods on a large number of data sets.

Weaknesses:

The paper makes a lot of assertions about human cognition that are questionable. For instance:
- "In essence, the process of classification is the process of concept cognition"
- "Concept contains our knowledge about the world, and we use concept to understand and organize the world. Without it, there will be no human intelligence at all."
- "Similarity (...) plays a crucial role in the process of human classification"
- "Concept is represented based on similarity for children, which is also a basic choice for adults"

Also, sometimes it is difficult to understand if the paper makes assertions about its own definitions or about human cognition as in "the intrinsic property of concept is just the fuzziness rather than the randomness".

I do not see a problem in using assumptions based on cognitive science for building models. In fact, most models in AI do that somehow. However, care should be taken to not state these assumptions in the paper as settled truths. I rather see the paper provide in advance a list of theories, hypotheses, and assumptions considered along with references for them, and then describe the model proposed using them as a basis.

Finally, without details on how well the other methods used for comparison were adjusted, it is hard to know if the comparison is fair.

---

> ### Author Response · Authors · 2022-08-01
> **Response to Reviewer iVDZ**
>
> General Response
>
> We thank reviewer for valuable comments. We carefully revised the manuscript according to these comments. Below is a point-by-point response to the comments.
>
> Response to Weaknesses
>
> 1 Concept consists of two parts, extent and intent, and they can induce each other. Concept cognition is to obtain the extent or intent of the concept. The process of classification is to match an object to a concept, that is, to get the extent of the concept. Therefore, the nature of classification is concept cognition in some sense.
>
> 2 Concept is the basic cognitive units of knowledge representation. And humans use concepts to organize and understand the world. And, concept is essential for humans intelligence (See detailed discussion in Chapter 1 of literature 13).
>
> 3 Similarity is the key of classification. (See detailed analysis in literature 13-15)
>
> 4 Related studies can be found in literature 15.
>
> 5 To ensure the fairness of comparison, literature 1 has made sufficient consideration and great efforts. The experimental results of all the comparison methods are from literature 1.
> In addition, the splits of training and test data sets follows the literature 1. And implementation details, hyper-parameter settings and optimization process settings of the proposed method are given in detail (see Appendix A.4).
> To sum up, as far as we known, the fairness of the comparison can be guaranteed.
>
> Response to Questions
>
> 1 Seen response to weaknesses 1.
>
> 2 In the manuscript, we do not make a distinction between "classification" and "categorization". We use “classification” in the whole manuscript, which is common in machine learning literatures.
> Given Definition 1, the possible ambiguity can be reduce greatly.
> The existing science researches still can not comprehensively illustrate it. But there is no doubt that supervised learning should be important for humans learning.
>
> 3 We have modified it into “DNN has exceed the human level in very specific settings” in revised manuscript.
>
> 4, 5 Up to now, it is still difficult to understand and explain how do humans classify. Therefore, a human-like classifier is also difficult to understand. And it is impossible to directly simulate the classification process of humans. We have modified it into “To implement a classifier that is easy to understand and interpretable, one effective approach is to draw on relevant research in cognitive science.”
>
> 6 These numbers are artificially assigned to illustrate the difference between fuzziness and randomness. We have modified the description to eliminate the misunderstanding.
>
> 7 We assume that the evaluator has access to the class labels of the test data. This is a common setting in machine learning.
>
> 8 It is a typo. We have modified it.
>
> 9 We have deleted this sentence to eliminate the misunderstanding.
>
> 10 The formula 1 is an abstract optimization problem for illustrating the core work mechanism of FLM. In NN-FLM, we add the regularization term to prevent overfitting. (See appendix A.4). To eliminate this misunderstanding, we have made corresponding modifications.
>
> 11 A direct solution is to select more exemplars for classes with higher intra-class variability.
> In the experiment, to simplify the experimental setting the number of exemplars of each class on all data sets is set as min(5, # training samples of the class). And the NN-FLM has achieved competitive results. If the number of exemplars is adjustable, the better performance should be obtained.
> In the modified version, the number of exemplars is set as an adjustable parameter.
> In addition, the reason why intra-class variation is high is probably because the features used for describing samples contain the information unrelated to the concept. For example, “car” and “trucks” belong to “vehicles”, but the differences between them are great. That is because the features used for describing “car” and “trucks ” contain the information unrelated to “vehicles”. If raw features are used directly to calculate the similarity, the similarity between “car” and “trucks” would be small. Therefore, the key to concept representation is the similarity, and the key to the similarity is the sample representation. In the learned representations, it is expected that the intrinsic information of the concept will be strengthened and the unrelated information will be weakened. This is also the design idea of NN-FLM.
> At last, the above discussions also demonstrate that similarity is the basis of concept representation.
>
> 12 You are right. We have modified it into “In this case, the class labels predicted by NN-FLM are in line with human cognition to a certain degree.”.
>
> 13 The experimental details are given in appendix A.4.
>
> Response to Typos and writing style
>
> 1-5 and 7-9 It has been modified. And we checked the manuscript carefully to avoid similar problems.
>
> 6 The inner brackets denote the elements of the Cartesian product, and the outer brackets denote the function.

---

### Meta-Review · Area_Chair_HAVy · 2022-08-26

**Recommendation:** Accept
**Confidence:** Less certain

**Metareview:**

The paper proposes an approach for the design of neural networks for classification based on fuzzy theory, and a specific implementation is presented and experimentally assessed. Arguments from cognition to justify the proposed approach are also used, although at the level of inspiration. The lack of reference to fuzzy systems based neural networks models in the relevant literature in the initial version of the paper has been solved in the revised version, and author's rebuttal seems to have clarified most of the issues raised by reviewers. The experimental assessment seems to be robust. Personally I find the jargon used in the paper a bit unfit for NeurIPS standards, however I do not think this should be a valid reason for rejecting a paper for which no serious drawback has emerged. In any case, I think it is good for NeurIPS to diversify the range of approaches and methodologies covered by the scientific program.

**Award:**

No

---

### Decision · Program_Chairs · 2022-09-14

Accept